# Enhancing Large Multi-Modal Auto-Regressive Models with Condition Contrastive Alignment

**Chendong Xiang**
Department of Computer Science
2024316344
xiangxyaw@gmail.com

**Mingdao Liu**
Department of Computer Science
2024316083
joshua720117@gmail.com

**Yuji Wang**
Department of Computer Science
2024310693
wangyuji24@mails.tsinghua.edu.cn

## Abstract

The rapid development of auto-regressive (AR) models in multi-modal generation has brought promising advancements, enabling coherent text, image, and video generation within a single framework. However, AR models still face significant challenges in practical application, especially in image generation where classifier-free guidance (CFG) is commonly used to enhance output quality. CFG, while effective, introduces substantial computational overhead and deviates from the simplicity of end-to-end auto-regressive generation. In this proposal, we aim to explore the potential of Condition Contrastive Alignment (CCA) within Emu3, a state-of-the-art multi-modal AR model, to address the reliance on CFG in image generation. By applying CCA, a recently proposed method for aligning AR models with target distributions through contrastive learning, we hypothesize that Emu3 can achieve comparable or superior output quality without CFG, reducing computational cost and improving generation efficiency. Our approach involves fine-tuning Emu3 with CCA on multi-modal data and conducting comprehensive evaluations across image and video generation benchmarks. This research will validate CCA's applicability to large AR models, potentially advancing the field towards more efficient, unified multi-modal generation frameworks.

## 1 Background and Related Works

**Multi-modal Models.** Multi-modal models [10, 13, 18, 21] have advanced rapidly in recent years, driven by the latest breakthroughs in language and vision models, particularly auto-regressive (AR) language models [1, 14] and diffusion-based visual generative models [2, 8, 12]. This line of research aims to develop models capable of handling multi-modal generation (*e.g.* text-to-image generation, text-to-video generation) and perception (*e.g.* vision-language understanding) tasks within a single framework. In this project, we focus on AR multi-modal models [19, 20], which are considered to have considerable potential due to the simplicity and scalability of auto-regressive methods. These models unify text, image and video data into discrete tokens, training and inference with the *next-token prediction* approach.

**Guided Sampling.** Although the training and inference of language and vision data can be unified through *next-token prediction* with auto-regressive models, there is still a gap in the sampling process of auto-regressive language and vision models. To enhance sample quality, visual generative

auto-regressive models rely on guided sampling methods [3, 6, 10, 12], which adjust the *sampling distribution* by modifying the sampling algorithm without fine-tuning the pre-trained model. Specifically, classifier-free guidance (CFG) masks the condition with a relatively low rate (*e.g.* 10%) during training, enabling the model to predict unconditional logits. Then, a combination of conditional and unconditional logits is used when sampling [9, 11]. CFG complicates the original training method of auto-regressive models (*i.e. next-token prediction*), and double the computational overhead of sampling. In contrast, auto-regressive language models leverage alignment fine-tuning based on Reinforcement Learning from Human Feedback (RLHF) to improve instruction-following abilities by adjusting the *model distribution* and keeping the sampling algorithm unchanged [1, 15]. Recently, Condition Contrastive Alignment (CCA) has been proposed to guide the sampling of visual generative auto-regressive models through a fine-tuning algorithm [4] derived from Noise Contrastive Estimation (NCE), providing an approach to unifying the sampling of auto-regressive language and vision models.

## 2 Proposed Method

**Problem Formulation.** Consider a sample (*e.g.* an image) $x$ represented by a sequence of $N$ discrete tokens $x = \{x_1, x_2, ..., x_N\}$. The probability of sample $x$ given condition $c$ (*e.g.* the description of the image) can be decomposed as:

$$p(x|c) = \prod_{n=1}^{N} p(x_n|x_{<n}, c) \tag{1}$$

Each token $x_n$ is conditioned only on $c$, which can also be represented as discrete tokens, and its previous input $x_{<n}$. An auto-regressive (AR) model $\theta$ learns the conditional probability $p_\theta(x_n|x_{<n}, c)$ and samples tokens one by one in generation.

**Review of CCA Method.** To enhance the sample quality under condition $c$, CCA [4] derives a fine-tuning method from guided sampling method and Noise Contrastive Estimation (NCE), where the loss is defined as

$$\mathcal{L}_\theta^{\text{CCA}} = -\mathbb{E}_{p(x,c)} \log \sigma \left[ \frac{1}{s} \log \frac{p_\theta(x|c)}{p_\phi(x|c)} \right] - \mathbb{E}_{p(x)p(c)} \log \sigma \left[ -\frac{1}{s} \log \frac{p_\theta(x|c)}{p_\phi(x|c)} \right]. \tag{2}$$

$p_\phi$ is a pre-trained AR model and is frozen during training. $p_\theta$ is the target model and is initialized from $p_\phi$. $s$ is the guidance scale. Intuitively, $\mathcal{L}_\theta^{\text{CCA}}$ maximizes the relative likelihood where the condition $c$ and the sample $x$ matches, and minimize the the relative likelihood where the condition $c$ and the sample $x$ are independent and most likely mismatch.

**Review of practical CCA Method.** To tractably sample from the joint distribution $p(x, c)$ and the product of two independent marginals $p(x)p(c)$, CCA [4] propose a practical training loss. Consider a training batch with $K$ samples $B = \{(x_k, c_k)^{1:K}\}$. A random permutation of $c_k$ in $B$ is used as samples from $p(x)p(c)$ and the original batch $B$ as samples from $p(x, c)$. Then the loss for fine-tuning is defined as

$$\mathcal{L}_\theta^{\text{CCA}} = -\log \sigma \left[ \beta \log \frac{p_\theta(x_k|c_k)}{p_\phi(x_k|c_k)} \right] - \lambda \log \sigma \left[ -\beta \log \frac{p_\theta(x_k|c_{\rho(k)})}{p_\phi(x_k|c_{\rho(k)})} \right] \tag{3}$$

where $\rho$ is a random permutation, $\beta$ and $\lambda$ are two hyper-parameters.

**Research Plan.** We plan to fine-tune Emu3 [20] with CCA method on multi-modal data and conduct comprehensive evaluations across visual generation benchmarks. Our preliminary proposal is to use JourneyDB [17], a dataset with over 4 million high-resolution images and corresponding annotations, as the fine-tuning dataset and to evaluate the generation performance with Fréchet Inception Distance [7] (FID) and Inception Score [16] (IS) on ImageNet [5] dataset. Considering the image resolution, fine-tuning iterations and model size, we estimate the fine-tuning process will require approximately one week on 8 NVIDIA A100 GPUs. This research will explore CCA's applicability to large AR models and more complex tasks, potentially advancing the field towards more efficient, unified multi-modal generation frameworks.

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
