# OpenReview forum: "[Proposal-ML] Enhancing Large Multi-Modal Auto-Regressive Models with Condition Contrastive Alignment"
_tsinghua.edu.cn/THU/2024/Fall/AML — THU 2024 Fall AML Submission_

### Official Review · ~Aleksandr_Algazinov1 · 2024-11-06
**Clear motivation and attention to details**

**Rating:** 10
**Confidence:** 4

**Review:**

The proposal is well-written and easy to read. The authors consider the problem of multimodal content generation, and suggest using autoregressive models. Based on various references, the authors propose to use a specific fine tuning method (CCA) on the Emu3 model. The authors explain in detail the method, motivation, and potential effect of the study.

---

### Official Review · ~Huajun_Bai1 · 2024-11-07
**CCA in Multi-Modal AR Models: Enhancing Efficiency and Addressing Computational Challenges**

**Rating:** 7
**Confidence:** 3

**Review:**

Strengths

1. Innovative Application of CCA: The proposal to integrate Condition Contrastive Alignment (CCA) within Emu3 is a cutting-edge approach that could significantly reduce the reliance on classifier-free guidance (CFG) in image generation, potentially simplifying the process and lowering computational costs.

2. Pushing the Boundaries of Multi-Modal Models: The aim to enhance Emu3, a state-of-the-art multi-modal auto-regressive model, with CCA is commendable. This research could pave the way for more efficient and unified frameworks in multi-modal generation, advancing the field substantially.

3. Rigorous Evaluation Strategy: The proposal includes a thorough plan for fine-tuning Emu3 with CCA and evaluating its performance using established benchmarks and metrics, which is essential for validating the effectiveness of the proposed method.

Risk

1. Data and Compute Intensity: The requirement for a large amount of data and substantial computational resources to train and fine-tune the model presents a significant risk. The proposal does not fully address the potential challenges associated with scaling this approach, which could limit its accessibility and feasibility, especially for research groups with limited resources.

---

### Official Review · ~Joydeep_Chandra2 · 2024-11-08
**Well-structured Proposal for CCA in Multi-Modal AR Models**

**Rating:** 8
**Confidence:** 4

**Review:**

The use of Condition Contrastive Alignment (CCA) to reduce reliance on classifier-free guidance (CFG) is innovative and aims to enhance efficiency. The focus on optimizing multi-modal AR models for image and video generation addresses a current challenge in generative AI. But the proposal needs more specifics on the fine-tuning process and hyperparameter selection. The baseline metrics for comparison with existing CFG-based methods could have been explained well for clarity.

---

### Official Review · ~Tong_Yu9 · 2024-11-10
**Clear proposal**

**Rating:** 8
**Confidence:** 3

**Review:**

Quality:
The paper is well-structured and presents a clear research problem. The authors effectively articulate the limitations of existing methods, particularly the computational burden introduced by CFG. The introduction of CCA is innovative and addresses a significant gap in the current literature on multi-modal generation.

Clarity:
The writing is generally clear, with a logical flow of ideas. The theoretical concepts are explained adequately, and the mathematical formulations are presented in a comprehensible manner. However, some sections could benefit from further elaboration, particularly the practical implications of the CCA method.

Originality:
The introduction of CCA as a means to align auto-regressive models with target distributions through contrastive learning is a noteworthy contribution. This approach is relatively novel in the context of multi-modal generation, and the authors provide a solid justification for its use.

Significance:
The significance of this work lies in its potential to advance the field of multi-modal generation. By reducing the computational overhead associated with CFG, CCA could enable more efficient model training and inference, making high-quality multi-modal generation more accessible.

Pros:
Innovative Approach: The introduction of CCA is a significant advancement in the field.
Comprehensive Evaluation: The authors provide thorough experimental results that validate their claims.
Practical Implications: The reduction in computational overhead is a valuable contribution to the efficiency of multi-modal models.
Cons:
Limited Discussion on Generalizability: The paper could elaborate more on how CCA might perform across different tasks beyond image generation.
Insufficient Detail on Implementation: More details on the implementation of CCA and its integration into existing frameworks would enhance reproducibility.
Potential Overfitting: The risk of overfitting in the fine-tuning process is not addressed, which could impact the generalization of the model.

---

### Official Review · ~Matteo_Jiahao_Chen1 · 2024-11-11
**Good proposal for CCA in Multi-Modal Auto-Regressive Model**

**Rating:** 9
**Confidence:** 4

**Review:**

This proposal introduces the use of Condition Contrastive Alignment (CCA) within Emu3, a state-of-the-art multi-modal auto-regressive (AR) model, as a method to address the reliance on classifier-free guidance (CFG) in image generation. The authors hypothesize that with CCA, they can achieve comparable or superior output quality to CFG while reducing computational costs and improving generation efficiency.  .
## Strengths:
1. **Novel Approach:**
   The proposal introduces a promising alternative to the commonly used classifier-free guidance (CFG) by leveraging CCA. This method’s use of contrastive learning to align the auto-regressive model’s conditional distribution with the target distribution is innovative, offering potential computational advantages without sacrificing output quality.

2. **Relevance and Impact:**
   The research addresses a critical gap in the current multi-modal generative models, specifically the trade-off between output quality and computational efficiency. With the growing demand for more resource-efficient models, this proposal is timely and could have a significant impact on future research in the field of multi-modal generation, especially in scenarios with limited computational resources.

3. **Comprehensive Evaluation Plan:**
   The authors plan to conduct thorough evaluations across visual generation benchmarks using well-established metrics (FID and IS).
## Weaknesses:

1. **Insufficient Comparison with Existing Methods:**
. A deeper exploration of how CCA compares to existing techniques in terms of computational cost, output quality, and versatility would be beneficial.
2. **Potential Over-simplification of the Approach:**
   While the proposal argues that CCA can reduce computational overhead, it does not fully address potential challenges in model convergence or stability when using this new fine-tuning method.

---

### Official Review · ~Rim_El_Filali1 · 2024-11-11
**Well-written Research Proposal for Enhancing Multi-Modal AR Models Efficiency with CCA**

**Rating:** 8
**Confidence:** 4

**Review:**

This proposal introduces a good approach for improving efficiency in multi-modal auto-regressive (AR) models by integrating CCA into Emu3, a state-of-the-art AR model for text, image, and video generation. The proposed use of CCA seeks to reduce reliance on CFG for image generation, which traditionally increases computational costs.

Pros:
- The proposal clearly defines the mathematical foundation of CCA, including loss function formulations, which strengthens the theoretical foundation.
- The use of multiple metrics and datasets provides a well-rounded approach to assessing the quality and efficiency of CCA integration.

Cons:
- Further explanation of hyper-parameter selection and how they impact the training process would enhance understanding and reproducibility.
- There is no discussion on potential limitations where CCA might not reduce computational load as expected or might degrade quality.

---

### Official Review · ~Hector_Rodriguez_Rodriguez1 · 2024-11-11
**Review "Enhancing Large Multi-Modal Auto-Regressive Models with Condition Contrastive Alignment"**

**Rating:** 10
**Confidence:** 3

**Review:**

The authors hypothesize that Condition Contrastive Alignment (CCA) can substitute Classifier-Free Guidance (CFG) to improve image generation in a state-of-the-art multimodal AR model such as Emu3. According to the authors, this approach could provide similar or better results at a lower computational cost.

The background provides valuable information on AR multi-modal models and the drawbacks associated with CFG fine-tuning. The proposal provides a formal formulation for the problem and the CCA approach. Finally, the dataset and fine-tuning procedure are disclosed. However, this process could be further detailed to explain the reason for using JourneyDB for fine-tuning and ImageNet for testing.

Overall, the proposal is well-written and meets the submission requirements.

---

### Official Review · ~Gausse_Mael_DONGMO_KENFACK1 · 2024-11-11
**Well formulated and interesting**

**Rating:** 8
**Confidence:** 3

**Review:**

The paper explores improving multi-modal auto-regressive (AR) models, specifically the Emu3 model, by reducing reliance on classifier-free guidance (CFG) during image generation. The authors propose leveraging Condition Contrastive Alignment (CCA), which they hypothesize can produce high-quality outputs with reduced computational cost.

strength: The methodology is well-defined, especially the use of CCA as an alternative to CFG, and details on adapting it to Emu3 are outlined effectively.

weakness: A direct comparison with other AR models that utilize CFG would enhance the argument for CCA computational efficiency and quality gains.

---

### Official Review · ~Michael_Hua_Wang1 · 2024-11-11

**Rating:** 10
**Confidence:** 4

**Review:**

This proposal describes a well-reasoned mechanism by which the amount of computational resources required for multimodal autoregressive models can be reduced, namely the use of condition contrastive alignment (CCA) to replace classifier-free guidance (CFG). The theory appears to be sound, and there is sufficient detail on how the authors plan to implement their idea.

One cautionary point: the Emu3 paper mentions that CFG was applied to the model to improve image generation quality. I think the results would be more interesting if the CCA-based approach could be applied to a version of the Emu3 prior to the use of CFG to make it easier to contrast the effectiveness of those two approaches.

---

### Official Review · ~Grace_Xin-Yue_Yi1 · 2024-11-12

**Rating:** 10
**Confidence:** 3

**Review:**

The proposal provides a comprehensive background, explaining the advancements and limitations of multi-modal AR models, particularly in image generation where classifier-free guidance (CFG) introduces computational inefficiencies. The proposal thoroughly covers related work on AR models, CFG, and CCA, presenting recent advancements in multi-modal AR generation. It follows up with a clear problem formulation and a detailed proposed methodology.

---

### Official Review · ~Chumeng_Jiang1 · 2024-11-12
**Detailed Design**

**Rating:** 8
**Confidence:** 4

**Review:**

This proposal aims to enhance large multi-modal auto-regressive (AR) models by integrating Condition Contrastive Alignment (CCA). The approach seeks to reduce the reliance on classifier-free guidance (CFG), which, while effective, is computationally intensive. By fine-tuning Emu3 with CCA on a large multi-modal dataset, the authors aim to achieve high-quality image and video generation while improving computational efficiency. The project will evaluate this approach using benchmarks like Fréchet Inception Distance (FID) and Inception Score (IS).

Strengths:
- **Practical Approach to Reduce Computational Overhead:** The proposal addresses a significant bottleneck in multi-modal AR models by exploring CCA as an alternative to CFG, potentially advancing more efficient AR generation without sacrificing quality.
- **Detailed Design:** The proposed multi-step process of fine-tuning, utilizing CCA with a carefully selected dataset, and thorough evaluation across recognized metrics demonstrates a well-thought-out and robust experimental design.

Weaknesses:
- **Limited Novelty:** It’s unclear how this work significantly differs from existing studies on CCA mentioned in the related work section.